# Effect of a Multi-Strain Probiotic on Growth Performance, Lipid Panel, Antioxidant Profile, and Immune Response in Andaman Local Piglets at Weaning

Gopal Sarkar [1] , Samiran Mondal [1] , Debasis Bhattacharya [2], Perumal Ponraj [2], Sneha Sawhney [2], Prokasananda Bala [2], Dibyendu Chakraborty [3], Jai Sunder [2] and Arun Kumar De [2,*]

1   Department of Veterinary Pathology, West Bengal University of Animal and Fishery Sciences, Kolkata 700037, India; gsarkar999@gmail.com (G.S.); vetsamiran@gmail.com (S.M.)

2   Animal Science Division, ICAR-Central Island Agricultural Research Institute, Port Blair 744101, India; debasis63@rediffmail.com (D.B.); perumalponraj@gmail.com (P.P.); snehasawhney88@gmail.com (S.S.); drprakashnrce@gmail.com (P.B.); jaisunder@rediffmail.com (J.S.)

3   Division of Animal Genetics and Breeding, Sher-e-Kashmir University of Agricultural Sciences and Technology of Jammu, R.S. Pura, Jammu 181102, India; dibyendu_vet40@yahoo.co.in

*   Correspondence: biotech.cari@gmail.com; Tel.: +91-9679515260

**Abstract:** This study aimed to investigate the role of a multi-strain probiotic compound containing *Bacillus mesentericus*, *Bacillus coagulans*, *Enterococcus faecalis*, and *Clostridium butyricum* as an in-feed zinc oxide (ZnO) alternative in growth performance, diarrhea incidence, antioxidant profile, lipid panel, stress, and immunity in piglets at weaning. Seventy-two piglets weaned at 27 ± 1 day were divided randomly into three groups with four replicates of six piglets each: (i) a negative control group (WC) fed only a basal diet, (ii) a probiotic group (WB) fed a basal diet with the current probiotic formulation, and (iii) a positive control (PC) group fed a basal diet with 2500 mg/kg ZnO. The experiment was conducted for 28 days. Probiotic supplementation showed a positive effect on growth performance and reduced the diarrhea rate. The mean body weight of the piglets in the WB and PC groups was significantly higher than that of piglets in the WC group (14.88 ± 0.12, 14.97 ± 0.13 vs. 13.80 ± 0.06 kg; $p \leq 0.001$). The addition of probiotic to the diet improved the lipid panel; the WB group showed a significantly higher level of high-density lipoprotein cholesterol (mg/dL) (32.67 ± 0.85 in WB vs. 12.48 ± 0.76 in WC; $p \leq 0.001$) and lower levels of total cholesterol (mg/dL) (59.78 ± 1.97 in WB vs. 119.11 ± 2.12 in WC; $p \leq 0.001$) and low-density lipoprotein cholesterol (mg/dL) (17.90 ± 1.12 in WB vs. 69.10 ± 3.37 in WC; $p \leq 0.001$) compared with the negative control group. Moreover, probiotic supplementation enhanced the antioxidant defense system and provided protection from oxidative damage by increasing the concentrations of serum catalase, glutathione-S-transferase, and superoxide dismutase and by decreasing the concentrations of serum malonyldialdehyde and total nitric oxide. Heat shock proteins and other stress markers, such as serum cortisol, were reduced in the probiotic-fed group. The probiotic group also displayed higher levels of serum IgG and IgM at all time points and higher IgA on day 28 compared with the negative control group. Altogether, these results indicate that feeding with the currently used multi-strain probiotic formulation minimizes weaning stress, thereby improving the growth performance, antioxidant profile, lipid panel, and systemic and mucosal immunity. Therefore, multi-strain probiotic compounds may be used to replace ZnO in weaned piglets.

**Keywords:** probiotics; pigs; growth performance; lipid profile; oxidative stress; cytokines; immunity

## 1. Introduction

In modern intensive farming systems, weaning is practiced to enhance the breeding efficiency of sows and economic profit of the farm [1]. In modern farms, it is general practice to wean piglets at three to four weeks of age [2]. Weaning, though a standard

management practice, is a stressful and traumatic event for piglets due to sudden dietary, social, and environmental changes [3]. Such periods of multiple stressors are linked to severe enteric infection, diarrhea, reduced feed conversion efficiency, loss of weight, and death in extreme cases [4,5], leading to enormous economic losses to the swine industry. At weaning, as the digestive capacity of a piglet is poor, enteric opportunistic pathogens residing in the gastrointestinal tract ferment undigested feed materials and generate toxic metabolites that damage the intestinal mucosa and ultimately result in diarrhea and the poor performance of the piglet [3,6]. Moreover, stressors associated with weaning disrupt or weaken the antioxidant defense system of the piglets, making them more prone to stress and infection [7]. Sometimes in commercial farming, piglets are weaned at only one or two weeks of age, which amplifies the detrimental effects of weaning [8]. Therefore, reducing weaning stress is extremely important and key to profitable pig farming.

The in-feed administration of antibiotics and zinc oxide (ZnO) has been widely used to combat post-weaning diarrhea and growth improvement in piglets [9,10]. However, the use of some antibiotics as growth promoters has been banned in the European Union, China, Japan, and Korea due to the increasing occurrence of antibiotic resistance in animals as well as in humans, the ultimate consumers of animal produce [11–13]. In recent years, concern regarding the use of ZnO has been raised, as the extensive use of ZnO is linked to environmental heavy metal contamination [10]. Moreover, the development of antibiotic-resistant microorganisms is a common consequence of the application of high doses of dietary ZnO [14,15]. Considering the negative consequences of in-feed ZnO administration to the environment and public health, the European Union recommended phasing out the medicinal application of ZnO in pig production by 2022 [16].

Consequently, in the recent past, much emphasis has been placed on finding a safe and practical alternative to ZnO to alleviate weaning stress and to maintain swine health and performance. Among the strategies that have been proposed, probiotic supplementation has proven to be an effective alternative to in-feed ZnO use because of its potential to stimulate the intestinal immune system, antioxidant status, nutrient digestibility, and potential to increase the production of antimicrobial peptides and cytokines in the intestinal tract [16–18]. Lactic acid bacteria (LAB), which include a variety of bacterial genera, like *Lactobacillus*, *Bacillus*, *Bifidobacterium*, *Streptococcus*, *Enterococcus*, and some other microbes, are the microorganisms used most frequently as probiotic agents [19]. Among these LAB, spore-forming Bacillus spp. have been considered the most promising as their spores can endure hostile environments and allow for extensive storage at room temperature [20].

The effectiveness of a probiotic is highly strain-specific; some strains provide more benefits to the host than others [21]. *Bacillus coagulans* is a well-studied and popular probiotic strain that has been approved by the European Food Safety Authority (EFAF) and the United States Food and Drug Administration (FDA) [22], and its beneficial effects on growth and health have been proven in weaned piglets [23]. The butyric acid-producing bacteria *Clostridium butyricum*, normally residing in the distal part of the large intestine, has been reported to improve the growth performance and enhance the barrier function of weaned piglets [24]. Another study reported that probiotics prepared with *Clostridium butyricum* and *Enterococcus faecalis* improved the growth performance, immune response, and gut health of weaned piglets [25]. *Enterococcus faecalis* is a frequently used probiotic strain that is a part of a healthy gut microbiota. Studies have proven that it can reduce weaning-induced villus atrophy and diarrhea and improve the growth performance of weaned piglets [26]. *Bacillus mesentericus* TO-A is another potential probiotic strain that, in combination with *Enterococcus faecalis* T-110 and *C. butyricum* TO-A, has improved the production performance and immune system of pregnant sows [27]. *B. coagulans* alone [23] or in combination with *E. faecalis* and *C. butyricum* [28] has been found to be beneficial to weaning piglets. However, the combined effect of *B. coagulans*, *E. faecalis*, *C. butyricum*, and *B. mesentericus* on weaning piglets has not been investigated extensively. This study was designed with the hypothesis that the multi-strain probiotic compound could be an alternative to ZnO administration in the alleviation of weaning stress in

piglets. Therefore, the objective of the current study was to investigate the combined effect of multi-strain probiotic supplementation including *B. mesentericus* TO-A, *B. coagulans* SNZ1969, *C. butyricum* TO-A, and *E. faecalis* T-110 on the growth performance, lipid metabolites, antioxidant defense system, and serum cytokine profiles of piglets at weaning in an island ecosystem.

## 2. Materials and Methods

### 2.1. Experimental Area

The present study was undertaken at the ICAR-CIARI institute pig farm, Port Blair, South Andaman, a district of the Andaman and Nicobar Islands (ANI). The ANI is an archipelago made up of 572 islands and islets, situated (lat. 6° to 14° north and lon. 92° to 94° east) in the intersection of the Andaman Sea (east side) and the Bay of Bengal (west side). It has a total surface area of 8249 sq. km and 1962 km of coastline.

### 2.2. Experimental Period

The present work was conducted during the months of January and February 2022. The highest and lowest air temperatures during the study period were 33.5 °C and 21.8 °C, respectively. During this period, the average relative humidity ranged from 64.0% to 79.5% and the Temperature Humidity Index (THI) varied from 75.6 to 81.5.

### 2.3. Experimental Animals

The experiment was conducted on indigenous Andaman local pigs (ALP), which are generally reared by the tribal farmers of these islands using a semi-intensive system of management. However, under intensive conditions, they can perform extremely well and attain a market weight of 65–70 kg at the age of 9 months. They provide livelihood and nutritional security to the farmers, particularly the tribal farmers of the ANI.

### 2.4. Source of the Probiotic

The probiotic (BIFILAC) used in the present study was procured from a commercial company (Tablets India Limited, Chennai, India). It is a multi-strain probiotic and contains four microorganisms; the details are presented in Table 1. The probiotic formulation was reported to be safe and claimed to offer no side effects [29].

**Table 1.** Composition of the probiotic.

| Microbial Composition | Strain Number | GenBank Accession Details | Deposition Details | Content per Gram of Product |
|---|---|---|---|---|
| *Enterococcus faecalis* | T-110 | AB687552: 16S DNA CP006030: Complete genome CP006031: Complete plasmid | 8936 * | $3 \times 10^7$ CFU/g |
| *Clostridium butyricum* | TOA | AB687551: 16S DNA CP014704: Chromosome 1 CP014705: Chromosome 2 CP014706: Plasmid | 8935 * | $2 \times 10^6$ CFU/g |
| *Bacillus mesentericus* | TOA | AB687550:16S DNA CP005997: Complete genome | 8934 * | $1 \times 10^6$ CFU/g |
| *Bacillus coagulans* | SNZ 1969 | KC146407: 16S DNA | MTCC 5724 | $5 \times 10^7$ CFU/g |

* All three strains were deposited with the international deposit agency in Japan.

### 2.5. Study Design

The study was conducted on seventy-two clinically healthy piglets (ALP, weaned at $27 \pm 1$ day) with an initial average body weight of $8.77 \pm 0.15$ (mean $\pm$ SD, kg). The piglets were randomly divided into three groups according to sex and body weight, with twenty-four animals in each group (six piglets per pen; four pens/replicates per treatment). Each pen ($1.5 \times 1.3$ sq. mt) with six animals (three males and three females) was considered as an experimental unit. Throughout the trial, all of the pens were provided with ad libitum feed and clean drinking water. The floor of the pens was made of concrete with a side wall of 0.7 m height to allow adequate natural ventilation. The grouping of the piglets was carried out as follows: (a) a weaned negative control group (WC), which received the basal diet without probiotics, (b) a positive control (PC) group, which received the basal diet with 2500 mg/kg ZnO [30], and (c) a weaned probiotic group (WB), which received the basal diet with probiotic supplementation (0.1% with feed, *E. faecalis* $3 \times 10^7$ CFU, *C. butyricum* $2 \times 10^6$ CFU, *B. mesentericus* $1 \times 10^6$ CFU, *B. coagulans* $5 \times 10^7$ CFU per kg of feed). The probiotic dose was standardized using a pilot study in which the above-mentioned dose was found to be the most effective in promoting growth and alleviating weaning stress. The composition of the basal diet is depicted in Table 2. Probiotics and ZnO were mixed with experimental diets using a feed mixer. The duration of the experiment was for 28 days.

**Table 2.** Composition of the basal diet and its nutritional levels.

| Ingredients | Percentage |
|---|---|
| Maize | 50.00 |
| Wheat bran | 15.00 |
| Soybean meal | 29.00 |
| Vitamin and trace min. mix | 2.50 |
| MCP | 1.00 |
| Salt | 1.00 |
| CaCO3 | 1.28 |
| DL-Methionine | 0.22 |
| Chemical formula of basal diet (on dry matter basis) | |
| Dry matter (%) | 91.46 |
| Crude Protein (CP) (%) | 20.72 |
| Crude fiber (%) | 3.490 |
| Ether extract (%) | 4.827 |
| Calcium (%) | 0.80 |
| Total phosphorus (%) | 0.63 |
| Lysine (%) | 0.80 |
| Methionine and cystine (%) | 0.70 |
| Metabolizable energy (ME) (kcal/kg) | 3382.31 |

The basal diet was prepared and given to the experimental piglets according to the nutritional guidelines of the National Research Council [31]. The nutrient composition values were analyzed through proximate analysis using standard methodologies. Vitamin and trace mineral mix has (per kg feed) vitamin D3 (4000 IU), vitamin K (16 mg), vitamin E (80 IU), vitamin A (20,000 IU), Ca-pantothenate (50 mg), niacin (120 mg), riboflavin (20 mg), pyridoxine (6 mg), thiamine (4 mg), folic acid (2 mg), vitamin B12 (0.08 mg), biotin (0.08 mg), Mn (73 mg), Cu Zn (56 mg), (15 mg), Co (0.5 mg), Se (0.4 mg), and I (0.3 mg).

### 2.6. Estimation of Diarrhea Rate in Piglets

For an estimation of diarrheal incidence, the feces of the piglets was examined visually every morning and afternoon. Scoring of the feces samples was carried out and the severity of diarrhea was evaluated as per the procedure recommended by Walsh et al. [32], where 1 = hard feces; 2 = slightly soft feces; 3 = soft, partially formed feces; 4 = loose, semi-liquid feces; and 5 = watery, mucous-like feces. Obtaining a fecal consistency score of 4 to 5 for

2 consecutive days was considered as diarrhea in the piglets. The diarrhea rate was then calculated [33].

### 2.7. Production Parameters

The individual body weights of the piglets from all groups were recorded at a weekly interval, i.e., at days 0, 7, 14, 21, and 28, with an electronic balance. Daily feed intake was recorded throughout the study period. Similarly, the production parameters like average daily feed intake (ADFI), average daily gain (ADG), and feed-to-gain ratio (F:G) were calculated on the basis of feed intake and daily body weight gain data.

### 2.8. Blood Sampling

From each piglet, a 10 mL blood sample was collected from the cranial vena cava and was placed into a vacutainer containing clot activator (Hebei Xinle Sci & Tech Co., Ltd., Shijiazhuang, China), following standard aseptic conditions. Sampling was conducted on day 0 and at weekly intervals thereafter up to the end of the experimental period. Serum separation was performed by keeping the vacutainer tubes at room temperature for 30 min, followed by centrifugation at $1200 \times g$ for 10 min at 4 °C. The serum samples were stored at $-80$ °C till further use.

### 2.9. Lipid Profile Analysis

Serum lipid profiles, including concentrations of the total cholesterol (TC), triglycerides (TG), and high-density lipoprotein cholesterol (HDLc), were determined using enzymatic methods with commercially available kits (Jeev Diagnostics Pvt. Ltd., Chennai, India; Spinreact, S.A., Spain, and Pathozyme Diagnostics, Kholapur, India, respectively). The concentration of LDLc was determined as follows: LDLc = TC − HDLc − (TG/5) [34]. Additionally, the cardiac risk factor (CRF = TC/HDLc) and atherogenic index (AI = (TC-HDLc)/HDLc) were also evaluated [35,36]. The lipid profile was measured on days 0, 7, 14, and 28; CRF and AI were calculated for days 14 and 28.

The total antioxidant activity (T-AOC) of the serum samples was assessed using a commercial kit obtained from HiMedia Laboratories (Mumbai, India). Serum T-AOC was detected with the reduction of the Cu (II)-chromogen complex to the Cu (I) complex, and absorbance was measured at 460 nm.

The serum activities of glutathione S-transferase (GSH), superoxide dismutase (SOD), and catalase were estimated using commercially available kits from the Cayman Chemical Company (Ann Arbor, MI, USA).

Serum malonyldialdehyde (MDA) levels were measured to determine the degree of lipid peroxidation [30]. The MDA concentration in the serum samples was estimated with 2-thiobarbituric acid, and the variations in absorbance were read at 534 nm.

T-AOC and MDA were measured at day 0 and then at weekly intervals throughout the study period, whereas the SOD, catalase, and GSH levels were evaluated on days 0, 7, 14, and 28.

### 2.10. Measurement of Stress Biomarkers

Total nitric oxide (TNO) and heat shock proteins (HSPs) were observed at day 0 and at weekly intervals till the end of the experiment, whereas the serum cortisol concentration was measured on days 0, 7, 14, and 28.

### 2.11. Antioxidant and Oxidative Profile
#### 2.11.1. Nitric Oxide Assay

The TNO level was analyzed using a commercially available NO estimation kit (HiMedia Laboratories, Mumbai, India).

### 2.11.2. Serum Cortisol Assay

The level of cortisol in serum was measured using a commercial cortisol detection kit from Arsh Biotech (Life Technologies, Delhi, India) using biotin double-antibody sandwich technology.

### 2.11.3. Determination of Serum HSPs

Serum heat shock proteins (HSP90, HSP70, HSP40, and HSP20) were determined using commercial double-antibody sandwich ELISA kits from Arsh Biotech (Life Technologies, Delhi, India).

### 2.12. Immune Parameters

Serum immunoglobulin concentrations, including IgM, IgG, and IgA, were determined using commercial kits from Arsh Biotech (Delhi, India).

Serum interleukin concentrations, including IL-1β, IL-2, IL-4, IL-6, IFN-γ, and IL-12, were determined using porcine ELISA-based kits from Arsh Biotech (Life Technologies, Delhi, India). All parameters were evaluated on days 0, 7, 14, and 28.

### 2.13. Statistical Analysis

Before data analysis, a Shapiro–Wilk statistics assay was performed to check the data for normality. The data had a homoscedastic distribution and had a normal shape. One-way repeated measures analysis of variance (ANOVA) or a within-subjects ANOVA was applied to determine the significant differences between groups at a particular time point in GraphPad Prism software (http://www.graphpad.com (accessed on 1 September 2023)). The pens served as the experimental units in the analysis of diarrhea rate, F:G ratio, and ADFI, while each pig was treated as the experimental unit for other experimental parameters of this study. The analyzed data for each parameter was presented as the mean (M) $\pm$ standard error of the mean (SEM). The statistical significance was defined as the mean values with a significance level of $p < 0.05$.

## 3. Results

### 3.1. Production Parameters and Diarrhea Incidence

The effect of the current probiotic formulation on the production parameters and diarrhea incidence is presented in Table 3. No significant difference in body weight was observed among the three groups at day 0. Thereafter, significantly higher body weights were observed in the WB and PC groups compared to the WC group on days 7, 14, 21, and 28, whereas no significant difference was observed between the WB and the PC groups. The WB and PC groups showed higher average daily weight gain (ADG) at 0–14 days and 14–28 days compared to the piglets of the WC group. Moreover, the overall ADG (0–28 days) of the WC group was significantly lower than the WB and PC groups. Significantly higher ADFI in the PC group compared to the WC group was recorded at 0–14 days, whereas it did not vary between the WB and PC groups. However, the overall ADFI (0–28 days) did not differ significantly among the groups. The WB and PC groups showed significantly lower feed-to-gain rations (F:G) compared to the control group (WC) for 0–14 days, 14–28 days, and overall (0–28 days). The diarrhea rates in piglets supplemented with the probiotic formulation (WB) or in-feed ZnO (PC) were lower than those of the negative control group (WC).

### 3.2. Lipid Panel Analysis

The level of TC did not vary significantly among the groups up to day 14 (Figure 1a). At day 28, the TC level was significantly higher in the piglets from the WC group than those from the other two groups (WB and PC), whereas no difference was recorded between the WB and PC groups.

**Table 3.** Production parameters and diarrhea incidence in piglets.

| Parameters | WC | WB | PC | *p*-Value |
|---|---|---|---|---|
| | | Body Weight/kg | | |
| 0 days | 8.84 ± 0.03 | 8.79 ± 0.04 | 8.75 ± 0.04 | 0.204 |
| 7 days | 9.87 [b] ± 0.09 | 10.11 [a] ± 0.07 | 10.38 [a] ± 0.05 | <0.001 |
| 14 days | 11.25 [b] ± 0.04 | 11.79 [a] ± 0.03 | 11.78 [a] ± 0.04 | <0.001 |
| 21 days | 12.58 [b] ± 0.05 | 13.63 [a] ± 0.10 | 13.71 [a] ± 0.11 | <0.001 |
| 28 days | 13.80 [b] ± 0.06 | 14.88 [a] ± 0.12 | 14.97 [a] ± 0.13 | <0.001 |
| | | ADG/g | | |
| 0–14 days | 171.90 [b] ± 3.09 | 214.17 [a] ± 4.08 | 216.19 [a] ± 4.61 | <0.001 |
| 14–28 days | 182.26 [b] ± 5.11 | 220.48 [a] ± 8.22 | 228.10 [a] ± 9.15 | <0.001 |
| 0–28 days | 177.08 [b] ± 2.08 | 217.32 [a] ± 4.49 | 222.14 [a] ± 4.84 | <0.001 |
| | | ADFI/g | | |
| 0–14 days | 417.73 [b] ± 7.51 | 454.03 [ab] ± 8.66 | 459.20 [a] ± 9.83 | 0.003 |
| 14–28 days | 466.59 ± 13.08 | 485.05 ± 18.08 | 499.20 ± 20.04 | 0.413 |
| 0–28 days | 442.16 ± 5.33 | 469.54 ± 9.81 | 479.20 ± 10.62 | 0.116 |
| | | F:G | | |
| 0–14 days | 2.43 [a] ± 0.01 | 2.11 [b] ± 0.01 | 2.12 [b] ± 0.004 | <0.001 |
| 14–28 days | 2.57 [a] ± 0.01 | 2.20 [b] ± 0.02 | 2.18 [b] ± 0.003 | <0.001 |
| 0–28 days | 2.49 [a] ± 0.02 | 2.16 [b] ± 0.04 | 2.16 [b] ± 0.003 | <0.001 |
| | | Diarrhea rate (%) | | |
| 0–28 days | 11.67 [a] ± 0.38 | 4.67 [b] ± 0.31 | 4.58 [b] ± 0.26 | <0.001 |

Analyzed data shown as M ± SEM. [a,b] Values with different superscripts in the same row differ significantly. WC: weaned negative control group, WB: weaned probiotic group, and PC: positive control (ZnO) group. Pen (n = 12, 4 numbers per group) was taken as an experimental unit in the analysis of ADFI, F:G, and diarrheal rate, while each pig (n = 72, 24 numbers per group) was treated as the experimental unit for body weight and ADG recording.

A downregulated HDLc concentration (Figure 1b) was observed in the WC group as compared to the WB and PC groups on days 14 and 28, whereas the HDLc concentration was higher in the WB group than the PC group on day 28.

On days 7 and 14, the TG concentration in the WC group was significantly greater than the other two groups (WB and PC), whereas it did not vary between the WB and PC groups (Figure 1c). However, day 28 recorded no significant difference in TG concentrations among the groups.

Regarding LDLc concentration (Figure 1d), a significantly higher value on day 28 was observed in the WC group compared to the other two groups, whereas between the WB and PC groups, the LDLc concentration was unchanged. At the other time points, there was no significant change among the groups in terms of LDLc concentration.

The WC group showed significantly higher values of CRF (Figure 1e) and AI (Figure 1f) on days 14 and 28 than the WB and PC groups.

### 3.3. Antioxidant Profiles and Oxidative Stress Indicators

Significantly lower T-AOC (Figure 2a) in the WC group was observed than in other two groups (WB, PC) throughout the study period, whereas no significant difference between the WB and PC groups was detected.

The activities of SOD (Figure 2b), catalase (Figure 2c), and concentrations of GSH (Figure 2d) in the PC and WB groups were significantly higher than those of the WC group on both days 14 and 28, whereas they did not vary between the two groups (WB and PC) except for SOD and catalase on day 7, in which both enzymes were found to be higher in the PC group than the WB group.

During the whole experimental period, the WC group showed significantly higher MDA concentrations than the WB and PC groups. However, no significant difference in MDA concentration was observed between the WB and PC groups throughout the study period, except on day 7, when the WB group showed significantly higher MDA concentrations than the PC group (Figure 2e).

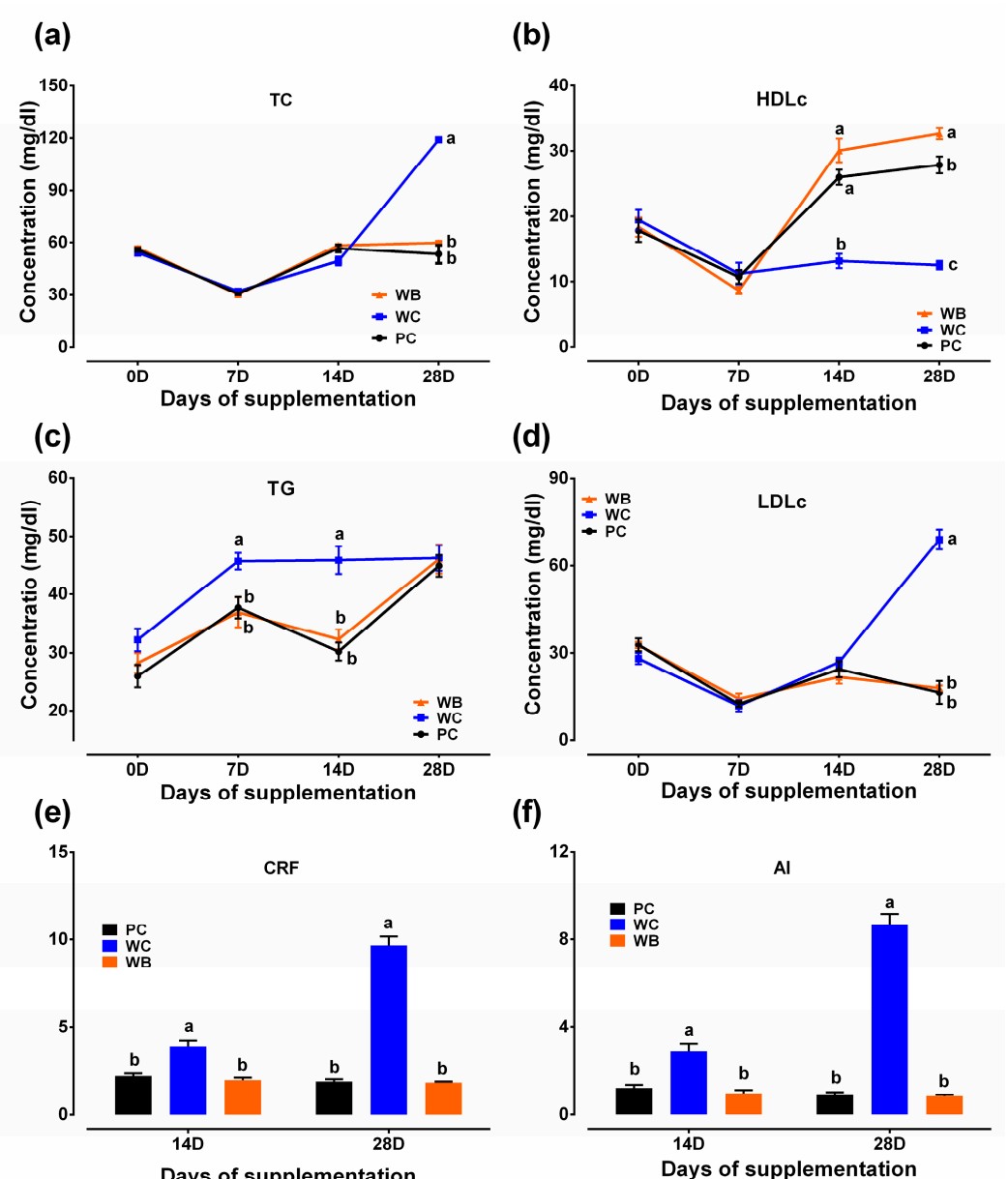

**Figure 1.** Effect on serum lipid panel of weaned piglets. (**a**) Total cholesterol (TC) concentration; (**b**) high-density lipoprotein cholesterol (HDLc) concentration; (**c**) triglyceride (TG) concentration; (**d**) low-density lipoprotein cholesterol (LDLc) concentration; (**e**) cardiac risk factor (CRF); (**f**) atherogenic index (AI). Analyzed data shown as M ± SEM. [a,b,c] Values at a particular time point with different superscripts differ significantly. WC: weaned negative control group, WB: weaned probiotic group, and PC: positive control (ZnO) group. Each pig (n = 72, 24 numbers per group) was treated as the experimental unit.

### 3.4. Evaluation of Stress Parameters

#### 3.4.1. Total Serum Nitric Oxide Concentration

The WC group showed significantly higher serum TNO concentrations (Figure 3a) on days 7 and 14 in comparison to the WC and PC groups, whereas there was no significant difference among the three groups on days 21 and 28.

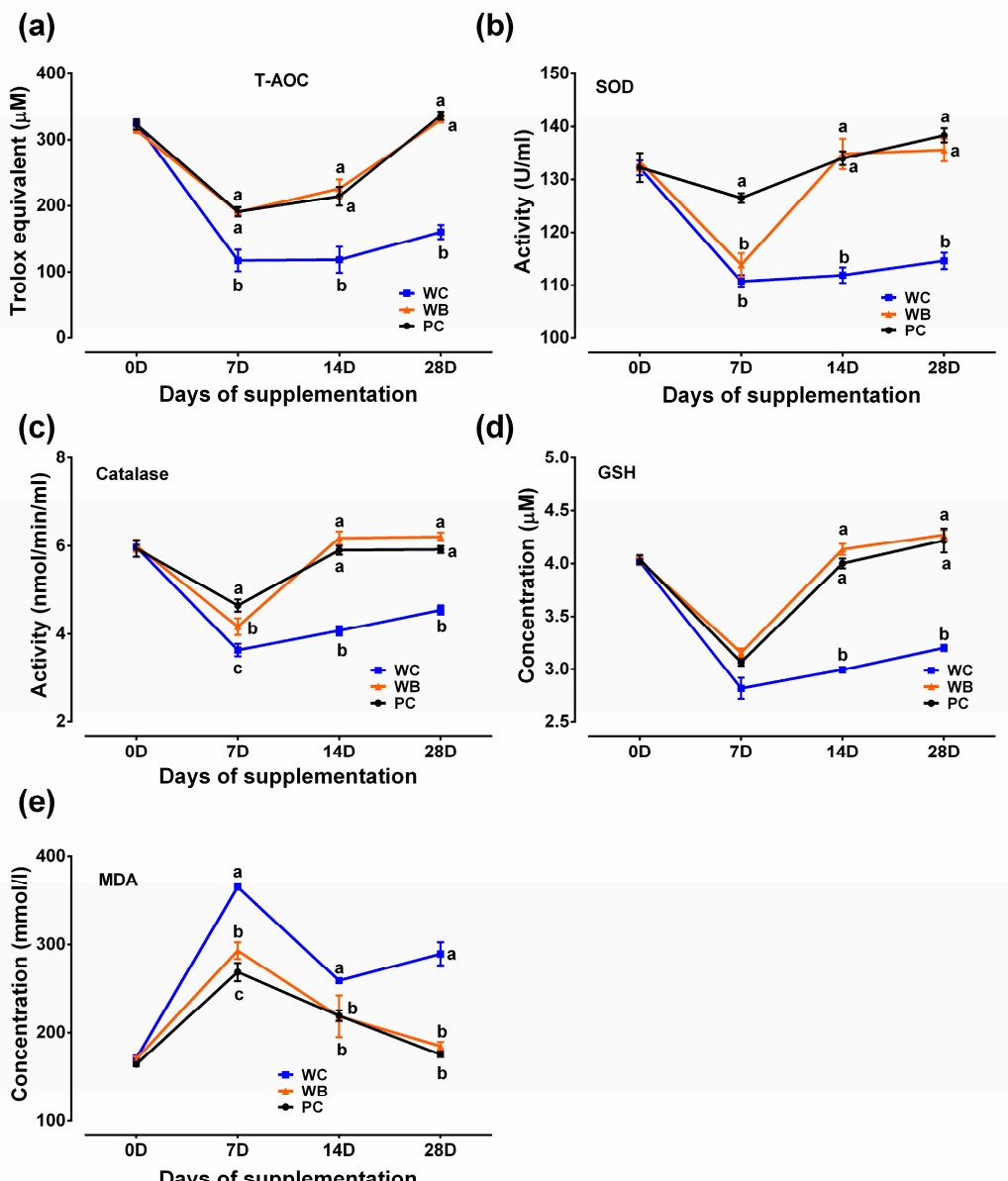

**Figure 2.** Results of probiotic administration on antioxidant activity and oxidative stress markers of weaned piglets. (**a**) Total antioxidant capacity (T-AOC); (**b**) superoxide dismutase (SOD) activity; (**c**) catalase activity; (**d**) glutathione S-transferase (GSH) concentration; (**e**) malonyldialdehyde (MDA) concentration. Analyzed data shown as M ± SEM. [a,b,c] Values at a particular time point with different superscripts differ significantly. WC: weaned negative control group, WB: weaned probiotic group, and PC: positive control (ZnO) group. Each pig (n = 72, 24 numbers per group) was treated as the experimental unit.

### 3.4.2. Serum Cortisol

Significantly higher cortisol concentrations (Figure 3b) in the WC and WB groups compared with that of the PC group were recorded on day 7. On days 14 and 28, significantly reduced concentrations of cortisol in the WB and PC groups were recorded compared to the WC group, while no significant difference was observed between the WB and PC groups on those days.

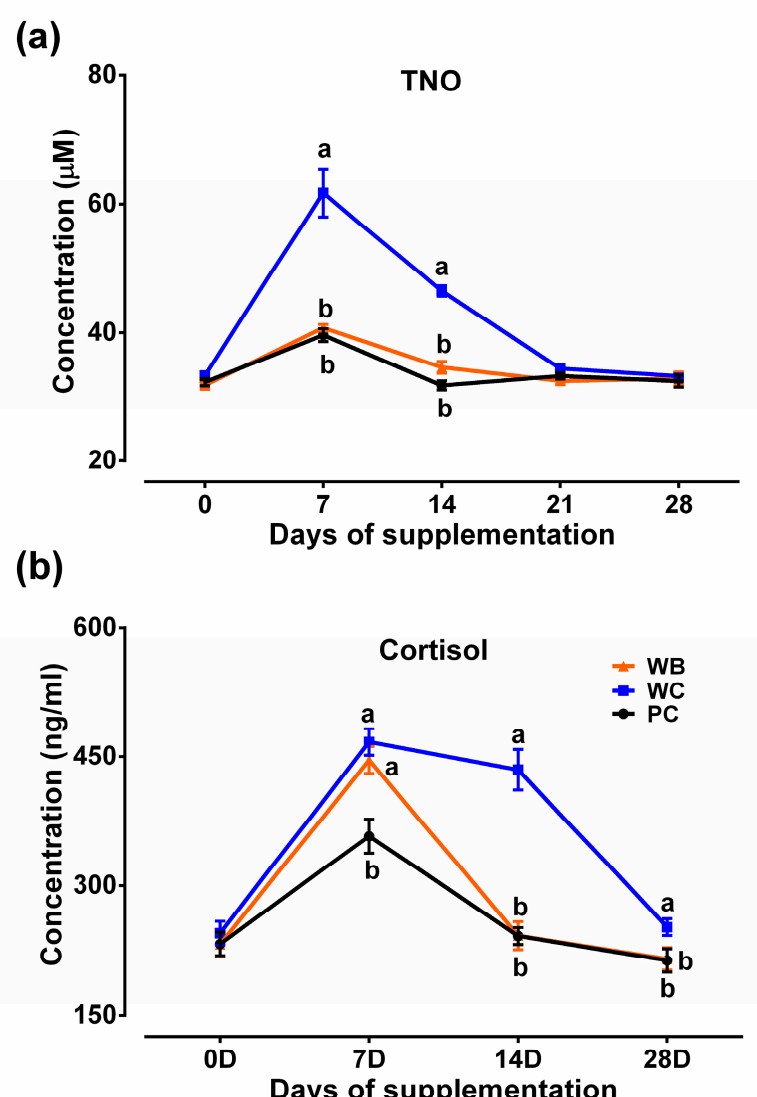

**Figure 3.** Results of probiotic supplementation on total nitric oxide (TNO) and cortisol concentration in weaned piglets. (**a**) Serum TNO level, (**b**) serum cortisol level. Analyzed data shown as M ± SEM. [a,b] Values at a particular time point with different superscripts differ significantly. WC: weaned negative control group, WB: weaned probiotic group, and PC: positive control (ZnO) group. Each pig (n = 72, 24 numbers per group) was treated as the experimental unit.

### 3.4.3. Serum Heat Shock Proteins (HSPs)

The serum concentrations of four HSPs levels (HSP20, HSP70, HSP40, and HSP90) were evaluated (Figure 4) in this study. Concentrations of all four HSP isoforms in the WC group were found to be upregulated at all time points compared to the WB and PC groups, except for HSP20 and HSP70 on day 7 and HSP90 on day 28. The HSP20 and HSP70 levels did not vary significantly between WC and WB at day 7. Concentrations of all four HSPs between the WB and PC groups were found to be insignificant at all time points.

### 3.5. Immune Parameters of Serum

The WB group recorded significantly greater levels of IgM and IgG on days 14 and 28 and IgA on day 28 than the other two groups, whereas the values in the WC group were found to be significantly lower than the other two groups at all time points for IgM and IgG and on day 7 for IgA (Figure 5a–c).

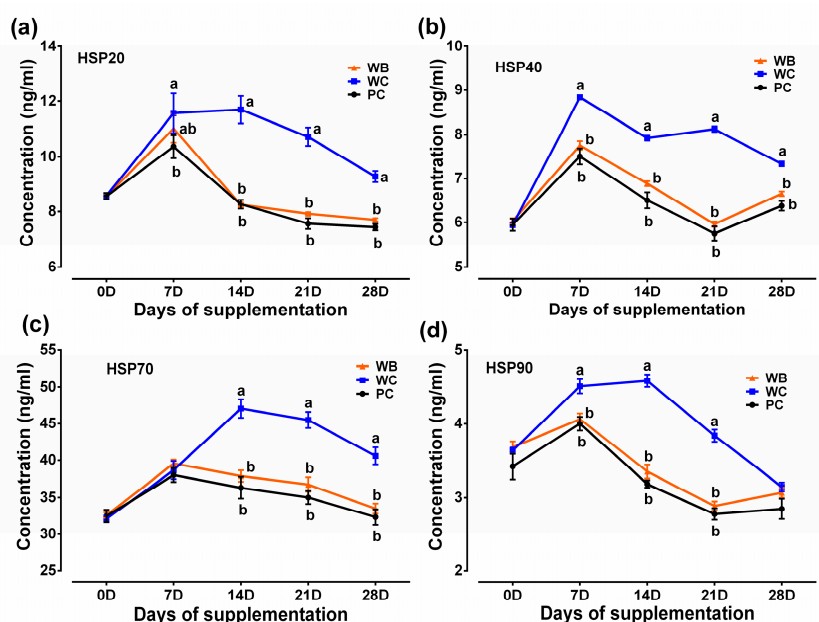

**Figure 4.** Results of probiotic supplementation on heat shock proteins (HSPs) in weaned piglets. (**a**) HSP20; (**b**) HSP40; (**c**) HSP70; (**d**) HSP90. Analyzed data shown as M ± SEM. [a,b] Values at a particular time point with different superscripts differ significantly. WC: weaned negative control group, WB: weaned probiotic group, and PC: positive control (ZnO) group. Each pig (n = 72, 24 numbers per group) was treated as the experimental unit.

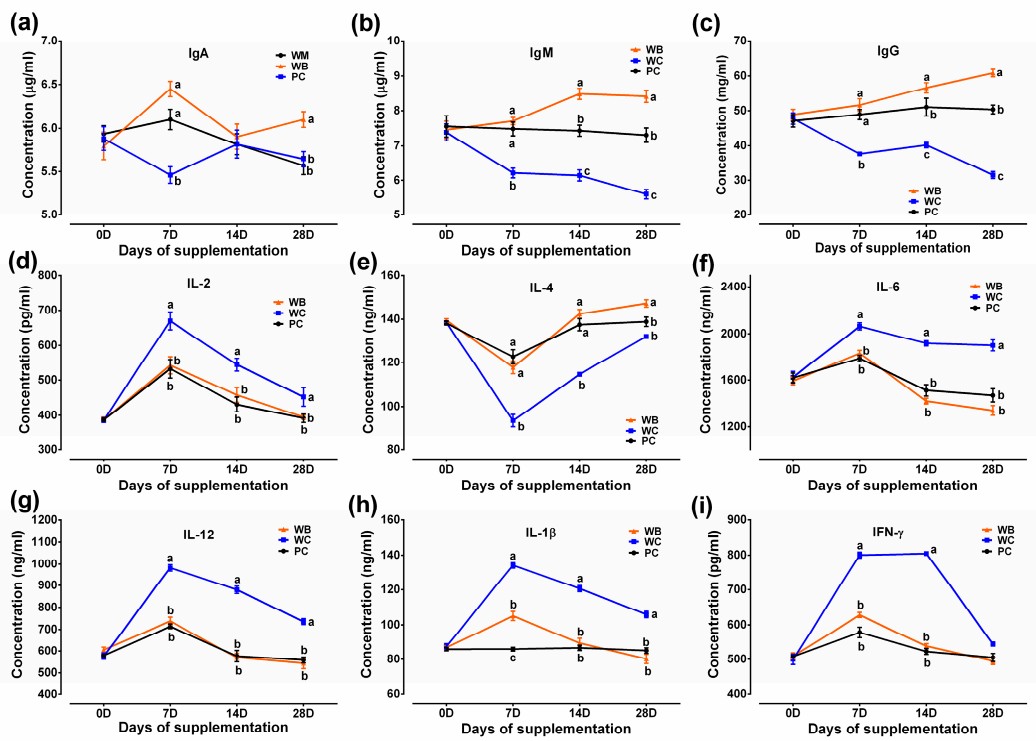

**Figure 5.** Results of probiotic supplementation on serum immunoglobulins and cytokines in weaned piglets. (**a**) IgA; (**b**) IgG; (**c**) IgM; (**d**) IL-2; (**e**) IL-4; (**f**) IL-6; (**g**) IL-12; (**h**) IL-1β; (**i**) IFN-γ. Analyzed data shown as M ± SEM. [a,b,c] Values at a particular time point with different superscripts differ significantly. WC: weaned negative control group, WB: weaned probiotic group, and PC: positive control (ZnO) group. Each pig (n = 72, 24 numbers per group) was treated as the experimental unit.

On days 14 and 28, IL-1β, IL-2, IL-6, IFN-γ, and IL-12 concentrations were found to be lower in the WB and PC groups compared with the WC group, while the levels did not vary between the WB and PC groups. The WC group showed reduced serum IL-4 concentrations at days 7 and 14 compared to the other two groups. At day 28, the WB group showed a significantly greater concentration of IL-4 than the other two groups (WC and PC) (Figure 5d–i).

## 4. Discussion

Weaning is the most stressful event in the life of a piglet [3]. Weaned piglets must adapt rapidly to the stressful condition to increase their growth and performance and to be healthy [4]. Probiotics can help weaning piglets adapt to this stressful condition by modulating the intestinal microbial population and stimulating the immune system of the host, which, in turn, can reduce diarrheal incidences and enhance gut health and growth performance [21,22,37]. Several bacterial species have been used as probiotics. Among them, lactic acid bacteria and butyric acid bacteria are commonly used in swine production systems [38]. In the present study, the multi-strain probiotic formulation containing *B. mesentericus* TO-A, *B. coagulans* SNZ1969, *C. butyricum* TO-A, and *E. faecalis* T-110 showed beneficial effects on the growth parameters, lipid profiles, antioxidant defense system, and immune parameters in weaning piglets; thus, it may be a good alternative to in-feed ZnO supplementation.

Body weight, ADFI, F:G ratio, and ADG are vital parameters for assessing animal performance in the pig industry. Here, the weaned piglets supplemented with probiotics (WB) demonstrated greater body weight, fortnightly ADG, and overall ADG compared with the weaned piglets of the negative control group, which were not supplemented with the probiotics, suggesting beneficial effects of the currently used multi-strain probiotic formulation. Moreover, compared with the negative control group, the probiotic-supplemented group had a significantly lower diarrhea rate as well as lower F:G ratio. However, the probiotic-supplemented group and positive control group did not show any significant difference in growth or incidence of diarrhea. Similar results were reported by Cai et al. [39], who observed improved growth performance in weaned piglets supplemented with *Bacillus*-based probiotics. Our findings were also well supported by a former study where weaned piglets that received a complex probiotic formulation (*Lactobacillussparacasei*, *Enterococcus faecium*, and *Bacillus subtilis)* had greater body weight gain, improved ADG, and a lower F:G ratio than the negative control piglets [40]. Probiotics are known to produce various types of enzymes, including arabinose, α-amylase, maltase, cellulase, levansucrase, dextranase, alkaline protease, β-glucanas, and neutral protease, which enhance the nutrient digestibility in the gut [41,42]. However, there are reports suggesting that the results of probiotic supplementation in the normal ration are inconsistent in nature. Giang et al. [43] and Méndez-Palacios et al. [44] reported that the inclusion of *Bacillus*-based or *Lactobacillus*-based feed additives failed to enhance the growth and production parameters of newly weaned piglets. The outcome of these studies may have been altered due to several other important factors, like the composition of the diet, form of the feed, interactions with probiotics, probiotic strains, probiotics doses, the age of the pigs, the surrounding environment, and the strategies of probiotic supplementation [45].

At weaning, piglets experience fasting due to the abrupt change in feed; this stimulates fat mobilization from their body reservoirs to support energy deficits [46]. High cholesterol concentrations, especially high concentrations of LDL, and high levels of TG in the bloodstream are closely related to atherosclerotic cardiovascular diseases [47,48]. In the present study, the probiotic formulation improved the lipid profile of the WB group, which had significantly greater HDLc and lower TC and LDLc concentrations than the WC group. Our observations were in accordance with previous studies. Yu et al. [49] found that selenium in combination with complex probiotics (*Lactobacillus acidophilus*, *Lactobacillus pentose*, and *B. subtilis*) tended to reduce TC, VLDL, and TG and increase HDLc concentrations in the blood. Kim et al. [50] reported that the cholesterol concentration was significantly de-

creased in pigs fed with *Lactobacillus*-based probiotic formulations. From this, it may be inferred that probiotic supplementation in weaning piglets can improve the lipid profile by decreasing the TC and LDL concentrations and increasing the HDLc concentration in serum.

Weaning stress often induces oxidative stress in animals [10,51]. In normal cellular metabolism, NO synthase (NOS) and NAD(P)H oxidase isoforms generate reactive oxygen species (ROS) and reactive nitrogen species (RNS), respectively [52]. At low or moderate concentrations, ROS and RNS are involved in a variety of physiological roles, including cell signaling pathways and mitogenic responses [53]. But, when ROS and RNS concentrations exceed the normal cellular level, they cause potential damage to essential biomolecules (DNA, proteins, and lipids), initiating a free radical chain reaction [54]. Therefore, oxidative stress and the subsequent over-production of ROS and RNS can reduce the immune response of the host, which, in turn, increase susceptibility to the pathogenic microorganisms, induce enterocyte apoptosis with cell cycle arrest in the gastrointestinal tract, and eventually decrease production performance [10,55]. In the current study, it was observed that the serum TNO concentration between the WB and PC groups did not vary throughout the experimental period. However, the TNO concentration was observed to increase significantly on days 7 and 14 in the WC group when compared with the other two groups. Furthermore, the MDA concentration was determined to investigate whether the weaning stress could lead to oxidative cellular damage. Cell membranes or plasma membranes are rich in polyunsaturated fatty acids that make them susceptible to free radical assault because of their multiple double bonds [56]. As a result of this oxidation of lipid molecules, MDA is produced, which interacts with the biomolecules and exerts cytotoxic and genotoxic effects [57]. So, MDA is used as a biomarker of oxidative damage [58]. In this study, the WB group recorded a lower serum MDA concentration than the WC group. These results suggest that the probiotic supplementation can reduce lipid peroxidation and oxidative damage to the cells. Similar findings were also observed in earlier reports [59,60].

A decrease in the total NO and MDA concentrations in probiotic-supplemented piglets over un-supplemented piglets indicated an improvement in the antioxidant system. The improved serum T-AOC offered additional support for an improvement in the antioxidant defense system in the probiotic-treated piglets. The antioxidant defense system in the body comprises several antioxidant enzymes such as superoxide dismutase (SOD), glutathione peroxidase (GSH-Px), and catalase [61]. SOD converts superoxide radicals to less-toxic $H_2O_2$ and then it is further processed to non-toxic water, either by GSH-Px or catalase [62]. Probiotics can enhance T-AOC by increasing the production of these antioxidant enzymes [63]. In the present study, the WB group showed elevated serum levels of T-AOC, SOD, catalase, and GSH on days 14 and 28 compared with the WC group in the same period. The increased T-AOC and antioxidant enzyme levels in the serum of the probiotic-supplemented group further supported the idea that the currently used probiotic formulation may contribute to an improvement in the antioxidant profiles of weaned piglets. Similar observations were also reported earlier [59,60]. LAB can degrade free radicals by producing intracellular enzymatic (SOD and catalase) and non-enzymatic antioxidants (glutathione and thioredoxin) [64,65]. This explains how the current probiotics can improve the antioxidant status of weaned piglets.

Stresses associated with weaning, especially oxidative stress, are known to enhance the production of heat shock proteins (HSPs) [66]. HSPs are highly conserved intracellular proteins that are conserved across the species, but cell death or tissue injury caused by physiological stress may increase their release into serum [67]. HSPs are involved in a variety of physiological functions, including protein synthesis, homeostasis, improving the antioxidant defense system, protecting the gut epithelium from oxidative stress and inflammation, and inhibiting the apoptotic pathways [68]. In this study, the probiotic-supplemented piglets (WB) showed significantly decreased serum levels of HSPs (HSP70, HSP40, HSP90, and HSP20) compared to the un-supplemented group (WC). This indicates a downregulating effect of probiotic supplementation on serum HSP levels in the weaned

piglets. Weaning increases the formation of free radicals, which induces the production of HSPs [66,69]. So, the downregulating effect of probiotics on serum HSPs might be due to the decreased formation of free radicals, and an increase in the total antioxidant capability. Similar results were also reported by Gan et al. [70], in which selenium-enriched probiotic supplementation reduced the mRNA expression of HSPs in heat-stressed piglets.

Cortisol is an important biomarker of stress. Weaning is a stressful event for a piglet and most frequently causes a marked rise in cortisol levels [71]. Cortisol is assumed to be produced by the hypothalamic–pituitary–adrenal (HPA) axis [72]. Under stress conditions like weaning stress, the HPA axis is activated and increases the production of the cortisol hormone, which is a healthy adaptive response of the body to cope with the adverse situation [73]. Cortisol, up to a certain level, is beneficial to the host, but at a higher level, chronic persistence may have deleterious effects on productivity [74]. In this study, the serum cortisol concentration significantly decreased in the probiotic-supplemented group on days 14 and 28 compared to the negative control group, whereas no significant difference was observed with the positive control group. These findings clearly indicate that weaning stress enhances the production of the cortisol hormone, which can be downregulated by probiotic supplementation. These findings provide further evidence that dietary probiotic supplementation can be instrumental in alleviating weaning stress. Our findings were in accordance with earlier studies, where Burdick Sanchez et al. [75] reported that the serum cortisol concentration was reduced in pigs fed with an *L. acidophilus* fermentation product. Similarly, Wang et al. [76] observed that the dietary supplementation of *Lactobacillus fermentum* I5007 in weaned piglets could decrease the diquat-induced plasma cortisol level.

It is a well-known fact that probiotics have a crucial role in immune system modulation, where probiotics interact with gut microbiota, gut epithelial cells, and immune cells, which, in turn, stimulate the immune function and antibody production in the host [77,78]. IgAs are the predominant isotype expressed in all mucosal tissues and aid mucosal immunity. The most prevalent antibody, IgG, is present in blood and extracellular fluid and is known to play important roles in regulating the systemic immune response. IgM is the main antibody produced during the initial stage of triggering the antibody-mediated immune response and is the major component of natural antibodies [79]. In the present study, the probiotic-supplemented group displayed higher levels of serum IgG and IgM throughout the experimental period; however, higher IgA was observed on day 28 when compared with the negative control group. Compared with the positive control group, significantly higher serum levels of all Igs were detected in the probiotic-supplemented group at the end of the experiment. These findings were in accordance with earlier studies. Dong et al. [80] demonstrated that the serum IgM and IgA levels were improved in weaned piglets treated with a complex probiotic formulation containing *Lactobacillus plantarum* and *B. subtilis*. Dlamini et al. [81] also observed an increased trend in IgG levels in weaned piglets upon supplementation with a combined probiotic formulation. The present results suggest that the direct-fed complex probiotics enhanced both mucosal and humoral immunity in the weaned piglets. These significant increases may be due to the persistence of current probiotic bacteria in the intestinal tract, acting as an immune adjuvant for the humoral immune system and therefore stimulating antibody production and titer. Then, the concentrations of serum pro-inflammatory cytokines, including IL-2, IL-1$\beta$, IFN-$\gamma$, IL-6, and IL-12, and anti-inflammatory cytokine IL-4 were determined to assess the effects of weaning stress on the intestinal or systemic inflammatory responses. Cytokines are small, cell-signaling molecules that play major roles in immune and inflammatory responses and maintain the overall homeostasis of the body [82]. An excess production of cytokines, especially pro-inflammatory cytokines, has negative influence on the immune response and gut integrity of the host, which ultimately reduces growth performance [83,84]. There is an intricate balance between pro and anti-inflammatory cytokines; anti-inflammatory cytokines suppress the production of pro-inflammatory cytokines and, thus, protect against intestinal inflammation and maintain gut integrity [82]. Their harmony is therefore es-

sential for the host immunological and inflammatory responses. Abrupt changes in the dietary and environmental factors during weaning may lead to changes in the cytokine network and cause transient inflammation of the gut mucosa, which may disrupt the barrier function [85]. In the current study, compared to the un-supplemented group (WC), the concentrations of pro-inflammatory cytokines were lower in the probiotic-supplemented group. This was in agreement with a previous finding where a *Bacillus*-based compound probiotic formulation (*C. butyricum*, *B. subtilis*, and *B. licheniformis*) modulated the inflammatory process in weaned piglets by decreasing serum pro-inflammatory cytokines (IL-1β, IL-6, and TNF-α) [86]. In the case of IL-4, the anti-inflammatory cytokine was significantly increased on days 7, 14, and 28 in the probiotic-supplemented group compared with the levels in the negative control group. Similar results were published by Laskowska et al. [87], where the multi-microbial probiotic formulation "Bokashi" raised the serum IL-4 level in pregnant sows. The experimental data of this study reveal that the presently used multi-strain probiotic formulation can reduce the transient inflammation of the gut and improve intestinal barrier function, which eventually can contribute to the improved growth performance of weaned piglets.

## 5. Conclusions

It may be concluded that the supplementation of the multi-strain *Bacillus*-based probiotic formulation containing *B. mesentericus*, *B. coagulans*, *E. faecalis*, and *C. butyricum* minimized the weaning stress of the piglets, thereby improving the feed intake, body weight, antioxidant activity, lipid profile, systemic and mucosal immunity, and overall growth performance of the weaned piglets of the Andaman local pig breed under the hot and humid climatic conditions of the Andaman and Nicobar Islands. Therefore, the multi-strain probiotic compound may be an appropriate replacement for ZnO in weaned piglets of this breed. As weaning stress is a global issue, the applicability of the study may be further explored in terms of other pig breeds as well. The study also recommends further study to help understand the molecular mechanism of the current probiotics in achieving beneficial effects on weaning stress in piglets. Moreover, a "challenged study" with pathogenic bacteria may be required to assess the therapeutic role of the current probiotic combination.

**Author Contributions:** Conceptualization, A.K.D. and G.S.; methodology, G.S., S.S., S.M., P.P., J.S. and D.B.; software, A.K.D.; validation, D.B.; formal analysis, D.C. and A.K.D.; investigation, G.S., P.P., S.S., P.B. and A.K.D.; resources, A.K.D. and D.B.; data curation, A.K.D.; writing—original draft preparation, A.K.D.; writing—review and editing, J.S. and D.B.; visualization, S.M.; supervision, D.B.; project administration, A.K.D.; funding acquisition, A.K.D. All authors have read and agreed to the published version of the manuscript.

**Funding:** The current study was funded by the "All India Coordinated Research Project on Pig" (AICRP on Pig), Indian Council of Agricultural Research, New Delhi, India, with grant number AICRP-Pig/ICAR-CIARI and the National Bank for Agriculture and Rural Development (NABARD), Port Blair, Andaman and Nicobar Islands, India, with grant number NABARD/IDA in Piglets.

**Institutional Review Board Statement:** The animal study protocol was approved by the Institutional Ethics Committee of ICAR-Central Island Agricultural Research Institute (ICAR-CIARI), Port Blair, Andaman and Nicobar Islands, India (protocol code ICAR-CIARI/AS/23468 and date of approval 23 December 2021). Humane animal care was practiced throughout the study and every effort was made to minimize the suffering of the animals. All procedures were carried out in conformity with the relevant national regulations and guidelines.

**Data Availability Statement:** All data are available within the manuscript.

**Conflicts of Interest:** There are no competing interests amongst the authors. The study's design, data collection, analysis, and interpretation; the preparation of the manuscript; and the decision to publish the findings were all made independently of the funders.

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
