# Peer review of "Effect of a Multi-Strain Probiotic on Growth Performance, Lipid Panel, Antioxidant Profile, and Immune Response in Andaman Local Piglets at Weaning"

_fermentation, doi:10.3390/fermentation9110970_

Round 1

Reviewer 1 Report

Comments and Suggestions for Authors

The authors have prepared an original and well-organized study. However, some modifications are required:

Abstract:

1.       The abstract should contain some relevant values from the results of the paper.

Introduction:

2.       Lines 44 to 48: I suggest adding enterics due to the fact that not all Enterobacteriaceae are fermenters (e.g. Salmonella spp.) and to delete mainly Escherichia coli and Salmonella.

At weaning, as the digestive capacity of a piglet is poor, enterics opportunistic pathogens (mainly Escherichia coli and Salmonella) residing at the gastrointestinal tract, ferment undigested feed materials and generate toxic metabolites which damage the intestinal mucosa and ultimately results in diarrhea and poor performance of the piglet [3,6].

……Or you can also rephrase, but using Campylobacter spp. and Salmonella spp. instead of mainly Escherichia coli and Salmonella’……

3.       When referring to a microorganism in a paper, the author should use the complete name of the microorganism in the first mention, then the genus name can be shortened to just the capital letter.

4.       The general reason for the combination of B. mesentericus, B. coagulans, C. butyricum and E. faecalis is not well introduced in the introduction part, particularly when we talk about an indigenous local pig that is robust and can survive with a very low level of management.

Materials and Methods

The study design is well-written and allows reproducibility.

Please correct the lines:

112:  2.5. . Study design

118: water. the pens have concrete floors

Results

5.       In Table 3, significance should be identified by placing the letter A at the highest value. Also, since there are no two statistical models that are denoted by A and a, I recommend that you write them in lowercase letters. I recommend you add the Standard Error of the Mean instead of the Standard Deviation, and P value since is more relevant to sustain your results.

Discussion

6.       Well done and sustained the results. However, what are the limitations of the study?

Conclusion

7.       Well done and sustained the results.

8.       Although the author aims to use this combination as a substitute for ZnO, he should consider that EU countries have strict legislation regarding the use of ZnO due to the environmental implications. In this context, please explain if it is a broad study that can be applied anywhere or is it for local use?

Author Response

The authors have prepared an original and well-organized study. However, some modifications are required:

Response: We greatly appreciate your careful reading of our manuscript and the comments you have provided to improve it. Below please find responses to the specific comments that you made.

Abstract:

  1. The abstract should contain some relevant values from the results of the paper.

Response: As suggested, some relevant values from the results have been included in the abstract section.

Introduction:

  1. Lines 44 to 48: I suggest adding entericsdue to the fact that not all Enterobacteriaceae are fermenters (e.g. Salmonella spp.) and to delete “mainly Escherichia coli and Salmonella.

At weaning, as the digestive capacity of a piglet is poor, enterics opportunistic pathogens (mainly Escherichia coli and Salmonella) residing at the gastrointestinal tract, ferment undigested feed materials and generate toxic metabolites which damage the intestinal mucosa and ultimately results in diarrhea and poor performance of the piglet [3,6].

……Or you can also rephrase, but using Campylobacter spp. and Salmonella spp. instead of “mainly Escherichia coli and Salmonella’……

Response: The line has been rephrased.

  1. When referring to a microorganism in a paper, the author should use the complete name of the microorganism in the first mention, then the genus name can be shortened to just the capital letter.

 Response: Modified as suggested.

  1. The general reason for the combination of B. mesentericusB. coagulansC. butyricumand E. faecalis is not well introduced in the introduction part, particularly when we talk about an indigenous local pig that is robust and can survive with a very low level of management.

Response:  It has been included in the introduction section.

Materials and Methods:

The study design is well-written and allows reproducibility.

Please correct the lines:

112:   2.5.. Study design

Response: Corrected

118: water. The pens have concrete floors

Response: Corrected

Results:

  1. In Table 3, significance should be identified by placing the letter A at the highest value. Also, since there are no two statistical models that are denoted by A and a, I recommend that you write them in lowercase letters. I recommend you add the Standard Error of the Mean instead of the Standard Deviation, and P valuesince is more relevant to sustain your results.

Response: As suggested the Table has been modified. Similarly, all the figures have been modified.

Discussion:

  1. Well done and sustained the results. However, what are the limitations of the study?

Response: The limitations of the study have been included in the conclusion section.

Conclusion

  1. Well done and sustained the results.

 Response: We appreciate your valuable comment.

  1. Although the author aims to use this combination as a substitute for ZnO, he should consider that EU countries have strict legislation regarding the use of ZnO due to the environmental implications.In this context, please explain if it is a broad study that can be applied anywhere or is it for local use?

Response: As the mechanism of stress in piglets in weaning is similar, it is highly reasonable to assume that the study can be applied in any breeds anywhere. Still we have included a sentence ‘As weaning stress in a global problem, the applicability of the study may be further explored in other pig breeds” in the conclusion section. 

Reviewer 2 Report

Comments and Suggestions for Authors

General comments:

This article deals with a very interesting topic that always requieres more information. 

Title: I suggest including that this trial was performed on indgenous Adaman local pigs.  

The introduction is clear and relevant

The materials and methods section is well-detailed, although it could possibly be summarized. 

About the diet, in Table 2.   I suggest specifing whether nutritional composition values are calculated or analyzed. If it is possible, please include the fiber values. 

I respectfully suggest, including the count of prebiotic bacteria in the final feed mixture. 

In the tables, I recommend adding the number of experimental units used for each analyzed variable in the footnote

Four observations may be, a very small sample size to evaluate certain variables. It is up to the editors to decide whether to accept this sample size. 

The results and the discussion are presented clearly, however,  I suggest including in the discussion some aspects related to specific expression of the variables associated with the genetics of the pigs used on this experiment. 

I suggest being more specific on the conclusion, mentioning that this results apply only to the conditions and type of pigs used on this trial.  

Author Response

General comments:

This article deals with a very interesting topic that always requieres more information. 

Response: We greatly appreciate your careful reading of our manuscript and the comments you have provided to improve it. Below please find responses to the specific comments that you made.

Title: I suggest including that this trial was performed on indgenous Adaman local pigs. 

Response: Included as suggested.  

The introduction is clear and relevant

Response: Thank you so much.

The materials and methods section is well-detailed, although it could possibly be summarized.

Response: Thank you so much. Necessary modifications have been made.

 About the diet, in Table 2.   I suggest specifing whether nutritional composition values are calculated or analyzed. If it is possible, please include the fiber values.

Response: The ration composition was formulated manually and the nutrient composition values were analyzed through proximate analysis using standard methodologies. Fiber value was included.

 I respectfully suggest, including the count of prebiotic bacteria in the final feed mixture. 

Response: Included as suggested.

In the tables, I recommend adding the number of experimental units used for each analyzed variable in the footnote. Four observations may be, a very small sample size to evaluate certain variables. It is up to the editors to decide whether to accept this sample size.

Response: Included as suggested.  

The results and the discussion are presented clearly, however, I suggest including in the discussion some aspects related to specific expression of the variables associated with the genetics of the pigs used on this experiment. 

Response: There is no specific expression of the variables associated with our experimental Andaman Local pigs that affects our experiment. Similar results are also reported by Sun et al., (2022) and Wang et al., (2019) in crossbred Duroc x Landrace x Yorkshire; by wang et al., (2009) in crossbred Duroc x Large White x Landrace, by Jørgensen et al., (2016) in crossbred (Landrace x Large White) x Pietrain; by Haupenthal et al., (2020) in purebred Large White and crossbred Landrace x Large White piglets; by Tissopi et al., (2020) in Hampshire piglets.

I suggest being more specific on the conclusion, mentioning that this results apply only to the conditions and type of pigs used on this trial. 

Response: Modified as suggested.